# Mutational Activation of the NRF2 Pathway Upregulates Kynureninase Resulting in Tumor Immunosuppression and Poor Outcome in Lung Adenocarcinoma

**DOI:** 10.3390/cancers14102543

**Published:** 2022-05-21

**Authors:** Johannes F. Fahrmann, Ichidai Tanaka, Ehsan Irajizad, Xiangying Mao, Jennifer B. Dennison, Eunice Murage, Julian Casabar, Jeffrey Mayo, Qian Peng, Muge Celiktas, Jody V. Vykoukal, Soyoung Park, Ayumu Taguchi, Oliver Delgado, Satyendra C. Tripathi, Hiroyuki Katayama, Luisa Maren Solis Soto, Jaime Rodriguez-Canales, Carmen Behrens, Ignacio Wistuba, Samir Hanash, Edwin J. Ostrin

**Affiliations:** 1Department of Clinical Cancer Prevention, The University of Texas MD Anderson Cancer Center, 1515 Holcombe Blvd., Houston, TX 77030, USA; jffahrmann@mdanderson.org (J.F.F.); xmao2@mdanderson.org (X.M.); jbdennis@mdanderson.org (J.B.D.); enmurage@mdanderson.org (E.M.); jpcasabar@mdanderson.org (J.C.); mceliktas@mdanderson.org (M.C.); jvykouka@mdanderson.org (J.V.V.); snpark@mdanderson.org (S.P.); olidel8@gmail.com (O.D.); hkatayama1@mdanderson.org (H.K.); shanash@mdanderson.org (S.H.); 2Department of Respiratory Medicine, Nagoya University Graduate School of Medicine, Nagoya 464-8601, Japan; ichidai@med.nagoya-u.ac.jp; 3Department of Biostatistics, The University of Texas MD Anderson Cancer Center, 1515 Holcombe Blvd., Houston, TX 77030, USA; eirajizad@mdanderson.org; 4Department of General Internal Medicine, The University of Texas MD Anderson Cancer Center, 1515 Holcombe Blvd., Houston, TX 77030, USA; jmayo1@mdanderson.org (J.M.); qpeng@mdanderson.org (Q.P.); 5Division of Molecular Diagnostics, Aichi Cancer Center, Kanokoden, Chikusa-ku, Nagoya 464-8681, Japan; a.taguchi@aichi-cc.jp; 6All India Institute of Medicine Sciences, Nagpur 441108, India; sctripathi@aiimsnagpur.edu.in; 7Department of Translational Molecular Pathology, The University of Texas MD Anderson Cancer Center, 1515 Holcombe Blvd., Houston, TX 77030, USA; lmsolis@mdanderson.org (L.M.S.S.); rodrigja2@gmail.com (J.R.-C.); cbehrens@mdanderson.org (C.B.); iiwistuba@mdanderson.org (I.W.)

**Keywords:** NRF2, KEAP1, lung adenocarcinoma, metabolism, kynurenine pathway, kynureninase, immune suppression, prognostic marker

## Abstract

**Simple Summary:**

Activation of the Nuclear factor-erythroid factor 2-related factor 2 (NRF2) pathway through gain-of-function mutations or loss-of-function of its suppressor Kelch-like ECH-associated protein 1 (KEAP1) is frequent in lung cancer. NRF2 activation has also been reported to alter the tumor microenvironment. Proteomic profiles of 47 lung adenocarcinoma (LUAD) cell lines (11 *KEAP1* mutant and 36 *KEAP1* wild-type) revealed the tryptophan-kynurenine enzyme kynureninase (KYNU) as a top overexpressed protein associated with activated NRF2. Mechanistic studies demonstrated that NRF2 is a regulator of enzymatically functional KYNU in LUAD. Analysis of multiple independent gene expression datasets of human lung cancer and a LUAD tumor microarray demonstrated that elevated tumor KYNU expression was associated with immunosuppression, including potent induction of T-regulatory cells, increased levels of PD1 and PD-L1, and poorer overall survival. Our findings indicate a novel mechanism of NRF2 tumoral immunosuppression through upregulation of KYNU.

**Abstract:**

Activation of the NRF2 pathway through gain-of-function mutations or loss-of-function of its suppressor KEAP1 is a frequent finding in lung cancer. NRF2 activation has been reported to alter the tumor microenvironment. Here, we demonstrated that NRF2 alters tryptophan metabolism through the kynurenine pathway that is associated with a tumor-promoting, immune suppressed microenvironment. Specifically, proteomic profiles of 47 lung adenocarcinoma (LUAD) cell lines (11 *KEAP1* mutant and 36 *KEAP1* wild-type) revealed the tryptophan-kynurenine enzyme kynureninase (KYNU) as a top overexpressed protein associated with activated NRF2. The siRNA-mediated knockdown of *NFE2L2*, the gene encoding for NRF2, or activation of the NRF2 pathway through siRNA-mediated knockdown of *KEAP1* or via chemical induction with the NRF2-activator CDDO-Me confirmed that NRF2 is a regulator of KYNU expression in LUAD. Metabolomic analyses confirmed KYNU to be enzymatically functional. Analysis of multiple independent gene expression datasets of LUAD, as well as a LUAD tumor microarray demonstrated that elevated KYNU was associated with immunosuppression, including potent induction of T-regulatory cells, increased levels of PD1 and PD-L1, and resulted in poorer survival. Our findings indicate a novel mechanism of NRF2 tumoral immunosuppression through upregulation of KYNU.

## 1. Introduction

Activation of the nuclear factor erythroid 2-related factor 2 (NRF2) pathway, either through gain-of-function mutation or loss-of-function of its suppressor, Kelch-like ECH-associated protein 1 (KEAP1), is one of the most dysregulated pathways in lung adenocarcinoma (LUAD) [1,2]. NRF2 is a critical stress response mediator in mammalian cells. NRF2 is regulated by KEAP1, which binds the N-terminal Neh2 regulatory domain of NRF2, mediating its degradation via polyubiquitination and thereby inhibiting NRF2 nuclear translocation and subsequent target gene expression [3]. Nuclear translocation of NRF2 due to loss of KEAP1 expression by biallelic inactivation of the gene via mutation, loss of heterozygosity or promoter methylation has been shown to frequently occur in *KRAS* mutant lung adenocarcinoma [2]. NRF2 regulates basal and inducible expression of hundreds of genes that contain antioxidant response elements (AREs) in their regulatory regions by heterodimerizing with small MAF proteins. NRF2-target genes are involved in multiple cellular pathways, including those that control reduction/oxidation (redox) homeostasis, drug metabolism and excretion, energetics, amino acid metabolism, iron metabolism, and mitochondrial physiology [4]. Activation of NRF2 has been associated with induction of chemoresistance and disease progression in several cancer types [5,6,7,8,9,10,11]. Recent evidence further demonstrates a pivotal role of NRF2 activation in modulating tumor metabolism and the tumor immunophenotype [12,13,14,15]. For example, NRF2 has been shown to regulate expression of key serine/glycine biosynthesis enzymes via activating transcription factor 4 (ATF4) to support glutathione and nucleotide production [15]. Prior studies have also shown that lung cancer cells harboring *KRAS* mutations reprograms cancer cell metabolite towards glutamine dependence through NRF2-mediated signaling activities that increase expression of enzymes involved in glutaminolysis [16,17,18]. Nevertheless, the extent to which NRF2 activation regulates tumor metabolism and how these changes impact tumor-immune interaction remains incomplete.

Here, we performed an initial screen of 47 lung adenocarcinoma (LUAD) cell lines to identify protein signatures related to *KEAP1* mutational status and activated NRF2, the results of which revealed the kynurenine-metabolizing enzyme kynureninase (KYNU) as a top differential overexpressed protein associated with activated NRF2. KYNU is a pyridoxal-5′-phosphate (pyridoxal-P)-dependent enzyme that catalyses the hydrolysis of kynurenine and 3-hydroxykynurenine into anthranilic and 3-hydroxyanthranilic acids, respectively. In mammalian cells, KYNU is involved in the biosynthesis of NAD cofactors from tryptophan through the kynurenine pathway [19]. Here, we show that the overexpression of KYNU was independent of induction of the entire tryptophan-kynurenine pathway (KP). We then evaluated the functional relevance of KYNU overexpression in LUAD and its impact on the tumor immunophenotype, showing marked immunmodulation towards a suppressive inflammatory infiltrate.

## 2. Materials and Methods

Detailed information regarding methodologies is provided in Appendix B.

### 2.1. Cell Culture and Transfection

Cancer cell lines were maintained in Roswell Park Memorial Institute (RPMI) media plus 10% fetal bovine serum (FBS) unless otherwise stated. Small interfering RNA (siRNA) transfection experiments were performed using the following siRNAs: siControl (Silencer Select Negative Control #1, Life Technologies, Carlsbad, CA, USA), siKYNU #1 and #2 (s17103 and s1704, Invitrogen, Waltham, MA, USA), si*KEAP1* #1 and #2 (#00080908 and #00344034, Sigma Aldrich, St. Louis, MO, USA) and si*NFE2L2* #1 and #2 (#00182393 and # 00341015, Sigma Alrich).

### 2.2. Chemicals

CDDO-Me (2-Cyano-3,12-dioxo-oleana-1,9(11)-dien-28-oic acid methyl, or bardoxolone methyl) was purchased from Sigma Aldrich. Stock solutions were resuspended in dimethyl sulfoxide (DMSO).

### 2.3. Western Blot Analysis

Primary antibodies include α-KYNU (Santa Cruz Biotechnology, Dallas, TX, USA, sc-390360; 1:200 dilution), α-IDO (Abcam, Cambridge, UK, ab55305; 1:500 dilution), α-QPRT (Abcam, ab171944; 1:1000 dilution), α-Nrf2 (Abcam, Cambridge, UK, ab62352; 1:500 dilution), α-KEAP1 (ProteinTech, Rosemont, IL, USA, 10503-2-AP; 1:1000 dilution), α-NQO1 (Abcam, Cambridge, UK, ab34173; 1:10,000 dilution), α-Peroxiredoxin-1 (Abcam, Cambridge, UK, ab41906; 1:10,000 dilution), and α-glutathione reductase (Abcam, Cambridge, UK, ab128933; 1:5000 dilution). ß-Actin primary antibody (Sigma Aldrich, St. Louis, MO, USA; 1:5000 dilution) was used as a control for protein loading. Uncut blots are provided in Figure A1 in the Appendix C.

### 2.4. RT-PCR Analysis

RNA was extracted using the RNeasy Extraction Kit (Qiagen, Germantown, MD, USA) according to manufacturer’s protocol. TaqMan PCR assay was performed with a 7500 Fast Real-Time PCR System using universal TaqMan PCR master mix (ThermoFisher, Waltham, MA, USA) and FAM^TM^-labeled probes for KYNU (Hs00187560_m1) and KEAP1 (Hs00202227_m1) and VIC^TM^-labeled probes for ß2M (Hs_00187842_m1). PCR was carried out using a BioRad CFX Connect RT System (Hercules, CA, USA). Values are reported as 2^−ΔΔCt^.

### 2.5. Proteomic Analysis

For proteomic analyses, each cancer cell line (*n* = 47 cell lines) was analyzed as a singular replicate. For these experiments, cancer cell lines (Table A1) were grown for seven passages in RPMI-1640 supplemented with ^13^C-lysine and 10% dialyzed FBS according to the standard SILAC protocol [20]. The purpose of SILAC labeling was to discriminate the FBS derived proteins which may affect the identification of the cell surface protein list.

### 2.6. Metabolomic Analysis

#### 2.6.1. Exometabolome Experiments

Exometabolome experiments were performed on conditioned media from 18 lung adenocarcinoma cell lines collected at predetermined incubation times (Baseline, 1, 2, 4 and 6 h) as previously described [21,22].

For si*KYNU* experiments, media (RPMI + 10% FBS) were collected 24 h post conditioning. Conditioned media was centrifuged at 2000× *g* for 10 min to remove residual debris and the supernatants were transferred and stored in −80 °C until use for metabolomics analysis.

#### 2.6.2. Assessment of KP-Related Metabolites

Metabolomics analysis for KP-related metabolites was conducted on Waters Acquity™ 2D/UPLC system (Milford, MA, USA) with parallel column regeneration configuration using H-class quaternary solvent manager and *I*-class binary solvent manager coupled to a Xevo G2-XS quadrupole time-of-flight (qTOF) mass spectrometer. Mass spectrometry data was acquired on Xevo G2 XS qTOF in ‘sensitivity’ mode for positive electrospray ionization modes within a 50–1200 Da range.

#### 2.6.3. Data Processing

Liquid chromatography mass spectrometry (LC-MS) and LC-tandem mass spectrometry (MSe) data were processed using Progenesis QI (Nonlinear, Waters, Milford, MA, USA) and values were reported as area units. Annotations for tryptophan, kynurenine, anthranilate and 3-hydroxyanthranilate were determined by matching accurate mass and retention times using authentic standards and by matching experimental tandem mass spectrometry data against the NIST MSMS or HMDB v3 theoretical fragmentations.

#### 2.6.4. Data Normalization

To correct for injection order drift, each feature was normalized using data from repeat injections of quality control samples collected every 10 injections throughout the run sequence. Measurement data were smoothed by Locally Weighted Scatterplot Smoothing (LOESS) signal correction (QC-RLSC), as previously described [22].

### 2.7. TCGA and Oncomine Datasets

Gene expression data, mutational information, and clinical data from The Cancer Genome Atlas (TCGA) network project was download from cBioPortal [23] (http://www.cbioportal.org/ (accessed on: 15 June 2021)). Gene expression data and associated clinical information for the Okayama Lung Study [24] was downloaded from the Oncomine database [25].

### 2.8. Immunohistochemistry

The immunohistochemistry used in this study comprised 124 surgically resected LUAD tumor specimens collected under an institutional review board protocol and archived as formalin-fixed, paraffin-embedded specimens in The University of Texas Specialized Program of Research Excellence thoracic tissue bank at The University of Texas MD Anderson Cancer Center. Patient characteristics for the analyzed cohort are provided in Table A2 in the Appendix C. Primary antibodies include KYNU (E-5, Santa Cruz, Santa Cruz, CA, USA, sc-390360, at 1:1000 dilution), PD-L1 (clone E1L3N, dilution 1:100; Cell Signaling Technology, Danvers, MA, USA), CD3 (T-cell lymphocytes; dilution 1:100; Dako), CD4 (helper T cell; Novocastra; clone 4B12, dilution 1:80; Leica Biosystems, Wetzlar, Germany), CD8 (cytotoxic T cell; clone CD8/144B, dilution 1:20; Thermo Fisher Scientific, Waltham, MA, USA), PD-1 (clone EPR4877-2, dilution 1:250; Abcam, Cambridge, UK), and FOXP3 (regulatory T cell; clone 206D, dilution 1:50; BioLegend, San Diego, CA, USA) [26].

Immunohistochemical expression of KYNU was evaluated in the cytoplasm of malignant cells using an H-score (% MCs with mild staining × 1 + % MCs with moderate staining × 2 + % MCs with strong staining × 3; range: 0–300). In this study, tumors which showed a >100 score for KYNU were considered positive for expression of the protein. The densities of cells expressing CD3, CD4, CD8, FOXP3, and PD-1 were evaluated using the Aperio nuclear algorithm and CD68 using Aperio cytoplasmic algorithm and counting the cells positive for them in five random square areas (1 mm^2^ each) in both intratumoral and peritumoral compartments as described elsewhere [26]. Histologic assessment of each 1 mm^2^ was performed to ensure that tumor tissue (at least 80% malignant cells and tumor stroma) was included in the selected intratumoral region, and only non-malignant cells were included in the peritumoral compartment. For this analysis, each area examined was overlapped with the sequential IHC slides to quantify each marker at the same location of the tumor specimens. The average total number of cells positive for each marker in the 5 square areas was expressed in density per mm^2^ [26]. Membranous PD-L1 expression in malignant epithelial cells and macrophages was analyzed using a cell membrane staining algorithm, and the staining intensity scored as 0 (no staining), 1+ (weak staining), 2+ (moderate staining), or 3+ (strong staining) and extension (percentage) of expression were determined. The PD-L1 H-scores for tumor tissues were determined by multiplying the staining intensity and reactivity extension values (range, 0–300).

### 2.9. Statistics

Statistical significance was determined using Kruskal–Wallis multiple comparison tests unless otherwise specified. Spearman correlation heatmaps, Cox proportional hazard models, and construction of Kaplan–Meier survival curves were carried out in R statistical software. Significance in survival distributions was determined by Mantel–Cox log-rank *t*-test. For survival analyses, we used the method described by Contal and O’Quigley [27] to derive an optimal change point for KYNU expression that yielded the largest difference between individuals in the two already defined groups (alive/dead) [22,28]. Variables included into the multivariable Cox proportional hazard models were based on the backward stepwise method (likelihood ratio).

Figures were generated in either GraphPad Prism v6 or R statistical software. We note that we chose to use quartiles in comparison of the bottom 25th and top 25th percentiles of KYNU mRNA expression with tumor immunophenotype as to highlight the effect between the most differential populations; Spearman correlation analyses based on the TCGA LUAD dataset using continuous variables for *KYNU* mRNA expression and gene signatures of tumor immune cell infiltrates.

## 3. Results

### 3.1. Analysis of the Protein Signature of KEAP1 Mutation in LUAD Cell Lines

We evaluated the proteomes of 47 LUAD (11 *KEAP1* mutant (mut) and 36 *KEAP1* wild-type (wt)) cell lines to identify protein signatures related to *KEAP1* mutation status (Table A1 in the Appendix C). Of the 3892 quantified proteins, 296 exhibited raw *p*-values < 0.05 (Figure 1A; Appendix A online). Differential analyses revealed kynureninase (KYNU), a downstream enzyme in the tryptophan-kynurenine pathway (KP), as one of the top overexpressed proteins in *KEAP1*mut LUAD cell lines (Figure 1A–C; Appendix A online). Other quantified enzymes in the KP were non-statistically significant between *KEAP1*mut and *KEAP1*wt LUAD cell lines, which includes the rate-limiting enzyme in *de novo* NAD+ biosynthesis, quinolinate phosphoribosyltransferase (QPRT), that is downstream of KYNU (Appendix A online). Analysis of gene expression data from the Cancer Cell Line Encyclopedia (CCLE) for LUAD yielded concordant findings at the RNA level (Figure A2A,B in the Appendix C). The overall findings suggest that the *KEAP1*mut-associated increase in KYNU expression is independent from induction of the entire kynurenine pathway, as would be observed with increased de novo NAD+ biosynthesis. Therefore, we focused our efforts toward the mechanism and biological consequence of KYNU upregulation in LUAD.

### 3.2. KYNU Protein Expression Is Regulated by NRF2 Activation in Lung Adenocarcinoma

*KRAS*, *EGFR*, *TP53*, *KEAP1*, and *STK11* are the most prevalent mutations in LUAD and mutations in *STK11* frequently co-occur with *KEAP1* mutations [29,30]. Stratification of LUAD cell lines based on the occurrence of these mutations revealed a positive association between elevated KYNU protein expression and *KRAS*mut LUAD cell lines that harbor mutations in *KEAP1* and *STK11* (Figure 1C; Figure A2C,D in the Appendix C). Analysis of The Cancer Genome Atlas (TCGA) LUAD gene expression dataset revealed a strong association between increased KYNU mRNA expression and occurrence of *KEAP1* mutations (Figure 1D; Figure A3A,B in the Appendix C).

To determine if NRF2 activation regulates KYNU expression, we performed siRNA-mediated knockdown of NRF2 in *KRAS/KEAP1/STK11* mutant LUAD cell lines H2030 and DFCI024, the results of which demonstrated reduced KYNU mRNA and protein expression (Figure 2A,B). No difference in KYNU protein expression was observed following siRNA-mediated knockdown of *STK11* (Figure A3C in the Appendix C).

Next, we activated NRF2 signaling in *KEAP1* wild-type/KYNU-low expressing LUAD cell lines H2009 (*KRASmut/KEAP1wt/STK11wt*), H1993 (*KRASwt/EGFRwt/KEAP1wt/STK11mut*) and H3255 (*EGFRmut/KEAP1wt/STK11wt*) via either siRNA-mediated knockdown of *KEAP1* or through chemical induction with the NRF2-activator CDDO-Me [31]. This led to increased protein and mRNA expression of KYNU (Figure 2C–F), implying that NRF2 could induce KYNU expression regardless of *KEAP1* mutation status.

To elucidate if KYNU was enzymatically functional, we performed metabolomic analyses using mass spectrometry on sequentially collected conditioned media (baseline, 30 min, 1 h, 2 h, 4 h and 6 h) from a subset of 18 LUAD cell lines. Analyses revealed a statistically significant positive correlation between the rate (area units/hour/100 μg of protein) of KYNU-derived anthranilate accumulation into conditioned media of LUAD cell lines with whole cell lysate KYNU protein concentration (Spearman *ρ* = 0.68 (95% CI 0.30–0.87); 2-sided *p*-value = 0.002) (Figure 2G,H, Table A3 in the Appendix C). Knockdown of *KYNU* in *KRAS/KEAP1/STK11* mutant LUAD cell lines H2030 and DFCI024 decreased media anthranilate (Figure 2I), thereby confirming that KYNU is enzymatically functional in LUAD cell lines.

### 3.3. Association between Tumor KYNU Expression and Tumor Immunophenotype

KP pathway-related metabolites are known to elicit immunosuppressive functions, particularly through inhibition of T-cell activation and promotion of regulatory T cells (Treg) differentiation [32,33,34,35]. As we observed only an induction of KYNU and not the entire KP pathway, we evaluated whether KYNU alone could phenocopy this immunosuppression. We first analyzed TCGA-LUAD mRNA expression datasets (see Methods) [36]. Spearman correlation analyses revealed statistically significant positive correlations between *KYNU* mRNA expression and gene-based signatures of several immune cell subtypes, including cytotoxic CD8+ T cells (*ρ* = 0.28 (95% CI 0.20–0.36); two-sided *p* < 0.0001) and Tregs (*ρ* = 0.18 (95% CI 0.10–0.27); two-sided *p* < 0.0001), as well as immune checkpoint blockade-related genes *CD247* (*ρ* = 0.29 (95% CI 0.20–0.37); two-sided *p* < 0.0001), *PDCD1* (*ρ* = 0.24 (95% CI 0.16–0.33); two-sided *p* < 0.0001), and *CTLA4* (*ρ* = 0.21 (95% CI 0.12–0.29); two-sided *p* < 0.0001) (Figure 3A, Figure A4 in the Appendix C, Appendix A online). We performed additional sub-analyses on TCGA LUAD tumors that were wild-type for *KEAP1*. This revealed that *KYNU* mRNA expression retained statistically significant positive correlations with the signatures of cytotoxic CD8+ T cells (*ρ* = 0.38 (95% CI 0.29–0.46); two-sided *p* < 0.0001) and Tregs (*ρ* = 0.30 (95% CI 0.21–0.39); two-sided *p* < 0.0001), and expression of *CD247* (PD-L1) (*ρ* = 0.32 (95% CI 0.23–0.41); two-sided *p* < 0.0001), *PDCD1* (PD-1) (*ρ* = 0.30 (95% CI 0.21–0.39); two-sided *p* < 0.0001), and *CTLA4* (*ρ* = 0.30 (95% CI 0.21–0.39); two-sided *p* < 0.0001) (Figure 3A, Appendix A online). LUAD tumors with *KYNU* mRNA levels in the top 25th percentile exhibited statistically significantly elevated levels (two-sided Wilcoxon rank sum test *p* < 0.01) signatures of CD8+ T cells and Tregs, and expression of *CD247* (PD-L1), *PDCD1* (PD-1), and *CTLA4*, as compared to those in the bottom 25th percentile (Figure 3B). Independent analysis of the Okayama [24] gene expression datasets yielded comparable findings (Figure A4 in the Appendix C).

Next, we performed immunohistochemistry for KYNU using a tissue microarray consisting of 124 LUAD tumors and compared KYNU protein levels (Figure 3C) with CD4+, CD8+, CD3+, and FOXP3+ immune-cell infiltrates, as well as PD-1 and PD-L1 staining positivity. Elevated KYNU protein expression was significantly positively associated with high numbers of CD8+ tumor infiltrating lymphocytes (TILs) (stratified by median, two-sided χ^2^ test for Trend *p* = 0.007), as well as PD-L1 cell positivity (stratified by median, two-sided χ^2^ test for Trend *p* = 0.02) (Figure 3D). No statistically significant association was observed between KYNU protein expression and staining positivity for CD3+, CD4+, FOXP3+, or PD-1. Collectively, these findings revealed that elevated KYNU expression is highly associated with an immunosuppressive tumor microenvironment.

### 3.4. Association between KYNU Tumor Expression and Overall Survival

We next evaluated the association between KYNU protein levels in a LUAD tissue microarray (TMA) and overall survival. In multivariate analyses, adjusted for stage, subjects with an optimal KYNU staining positivity cutoff of >120 exhibited statistically significantly worse overall survival compared to those subjects with KYNU staining positivity ≤ 120 (Hazard Ratio (HR) = 3.11 (95% CI 1.41–6.89), two-sided *p* = 0.005, Table 1, Kaplan–Meier survival curve in Figure 4A). A similar association between high *KYNU* expression and poorer overall survival was observed in LUAD gene expression datasets from Okayama lung study and TCGA (Figure 4B,C; Table A4 and Table A5 in the Appendix C. Thus, elevated tumor KYNU expression is an independent prognostic marker of poor overall survival in LUAD.

## 4. Discussion

While both activation of the NRF2 pathway and aberrant tryptophan catabolism are strongly linked to an immunosuppressive milieu in tumors, these pathways have not been directly linked [34,37,38,39,40]. Here, we demonstrate a novel finding of NRF2-mediated KYNU upregulation in lung adenocarcinoma that is prognostic for poor overall survival. Mechanistic studies revealed that KYNU is activated by NRF2 signaling and that KYNU overexpression is associated with an immunosuppressive tumor microenvironment characterized by elevated tumor T-cell infiltration, including T regulatory TILs, and concordant increases in protein expression of immune checkpoint blockade-related PD1 and PD-L1 (Figure 5).

A growing body of literature supports our observation that lung adenocarcinomas mutated for *KRAS* and with an activated NRF2 pathway have an altered tumor microenvironment that can, in part, be attributed to changes in tumor metabolism [16,17,18]. For instance, cancer cells harboring *KRAS* mutations have been shown to reprogram cancer cell metabolism towards an increase in uptake and catabolism of amino acids, such as glutamine and tryptophan, with pro-tumoral effects [41,42,43]. Reduced bioavailability of glutamine and tryptophan are reported to promote tumor immune suppression [35,44,45,46,47]. For example, depletion of tryptophan bioavailability in the tumor microenvironment triggers control non-derepressible 2 (GCN2)-mediated T-cell apoptosis [48] and attenuates T-cell proliferation, whereas increased accumulation of its downstream catabolite, kynurenine, promotes immune tolerance by inhibiting proliferation of T cells and natural killer cells and increasing proliferation of Tregs and myeloid derived suppressor cells [49,50]. These insights have thus led to several clinical trials specifically targeting tumor metabolism in KRAS/NRF2 tumors. For instance, the KEAPSAKE trial (Clinicaltrials.gov; NCT04265534) evaluated addition of a glutaminase inhibitor (telaglenastat) to standard-of-care immunotherapy and chemotherapy in advanced lung cancer. Similarly, intensive investigation into the kynurenine pathway has led to the development of several inhibitors including epacodostat and indoximod, which target the rate-limiting enzymes in tryptophan metabolism IDO1, IDO2, and TDO [51]. Despite early success, a recent phase III double-blinded randomized trial of epacadostat/pembrolizumab versus pembrolizumab (ECHO-301) did not note additional benefit of epacadostat [52]. Plausible explanations for lack of additional efficacy can be attributed to insufficient inhibition of tryptophan catabolism or due to disparity in expression of IDO and TDO amongst different cancer types [20]. Additionally, other tryptophan catabolites in the kynurenine pathway, such as 3-hydroxyanthranilate, can exert immunosuppressive functions by directly inhibiting T-lymphocyte activation, promoting regulatory T cell (Treg) differentiation, and in mitigating non-antigen stimulated T-cell proliferation [39,53]. There remains interest in development of pleiotropic tryptophan pathway inhibitors or a combination of inhibitors that act on multiple enzymes within the kynurenine pathway. Our findings reported are therefore of direct relevance, by identifying a potential alternative target to attenuate immunosuppression. Further investigations exploring the targetability of KYNU as a ‘immuno-metabolic’ adjuvant in LUAD harboring *KEAP1* mutations are thus warranted.

Elevated levels of circulating tryptophan and kynurenine-pathway related metabolites have also been reported to be associated with an increase in risk of developing lung cancer [54,55] and poor overall survival [56,57]. A recent report demonstrated that elevated the plasma levels of 3HA, a metabolite derived via the catabolism of 3-hydroxykynurenine by KYNU, was associated with significantly worse progression free survival in NSCLC subjects [58]. Notably, in this study, the combination of high tumor PDL-1 expression with elevated plasma 3HA had the highest predictive accuracy of objective response to immune checkpoint inhibitors (ICI). Here, we report that KYNU is an independent prognostic indicator of poor overall survival in lung adenocarcinoma with increased CD8+ and T-regulatory lymphocyte infiltration into tumors. It has been demonstrated that in tumor-bearing immune competent mice, administration of pharmacologically optimized PEGylated kynureninase (PEG-KYNase) promoted anti-cancer effects via increases in tumor infiltration and expansion of CD8+ lymphocytes [20]. Discrepancies may be attributed to our finding that KYNU is regulated through NRF2 activation, which is a known predictor of poor patient survival [17,59,60]. Additionally, lung cancer commonly occurs in a background of a chronically inflamed lung, which could suppress T-cell function through immune checkpoint blockade or through exhaustion. This may reflect the dynamic nature of T-cell activation and suppression in the tumor microenvironment.

## 5. Conclusions

We have identified a distinct signature of perturbed tryptophan catabolism in subsets of lung adenocarcinomas with activated NRF2 characterized by elevated KYNU expression. Protein expression of KYNU serves as a promising prognostic marker for lung adenocarcinoma and may yield capacity to identify subjects who are likely to receive benefit from ICI therapy. Further exploration of KYNU in the context of lung adenocarcinoma and immunotherapy is warranted.

## Figures and Tables

**Figure 1 cancers-14-02543-f001:**
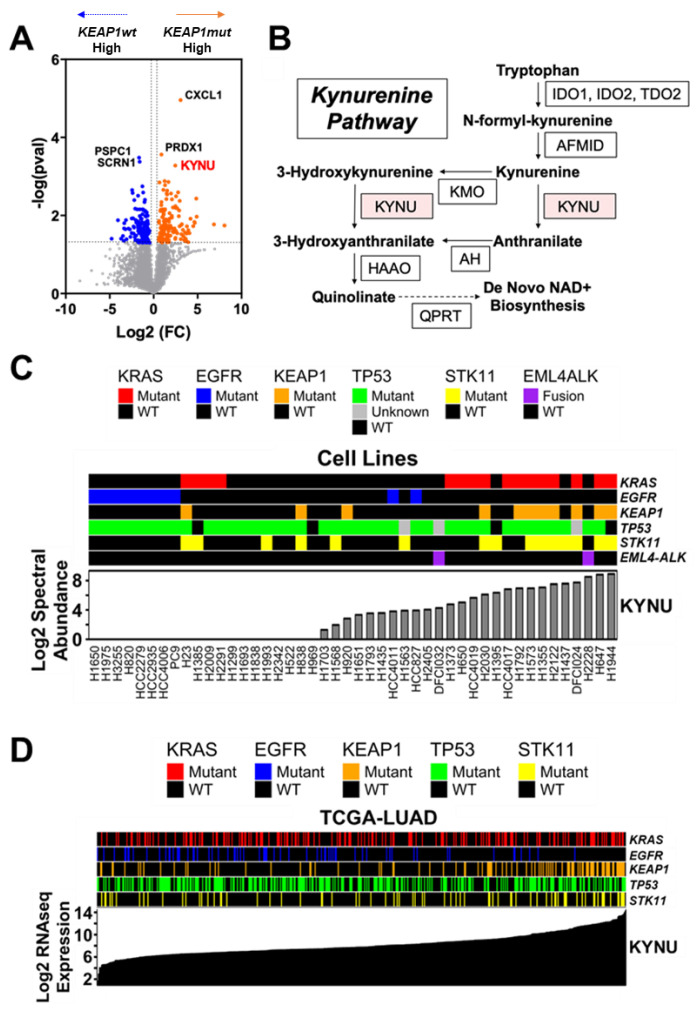
Lung Adenocarcinoma (LUAD) cell lines exhibit elevated KYNU expression. (**A**) Volcano plot illustrating differentially expressed proteins between *KEAP1mt* (*n* = 11) and *KEAP1wt* (*n* = 36) LUAD cell lines. Node color represent proteins that are statistically significantly (two-sided student *t*-test *p* < 0.05) increased (orange nodes) or decreased (blue nodes) in *KEAP1mut* LUAD cell lines. (**B**) Schematic of the kynurenine pathway (KP). (**C**) Association between whole cell lysate extract KYNU protein expression and presence of *KRAS*, *EGFR*, *KEAP1*, *TP53*, and *STK11* mutations and *EML4-ALK* fusions amongst 47 LUAD cell lines. KYNU expression is ranked from lowest to highest. (**D**) Association between KYNU mRNA expression and presence of *KRAS*, *EGFR*, *KEAP1*, *TP53*, and *STK11* mutations in TCGA-LUAD. KYNU mRNA expression is ranked from lowest to highest. Abbreviations: IDO—indoleamine 2,3-dioxygenase; TDO—tryptophan 2,3-dioxygenase; AFMID—kynurenine formamidase; KYNU—kynureninase; KMO—kynurenine 3-monooxygenase; AADAT—aminoadipate aminotransferase; HAAO—3-hydroxyanthranilate 3,4-dioxygenase; QPRT—quinolinate phosphoribosyltransferase.

**Figure 2 cancers-14-02543-f002:**
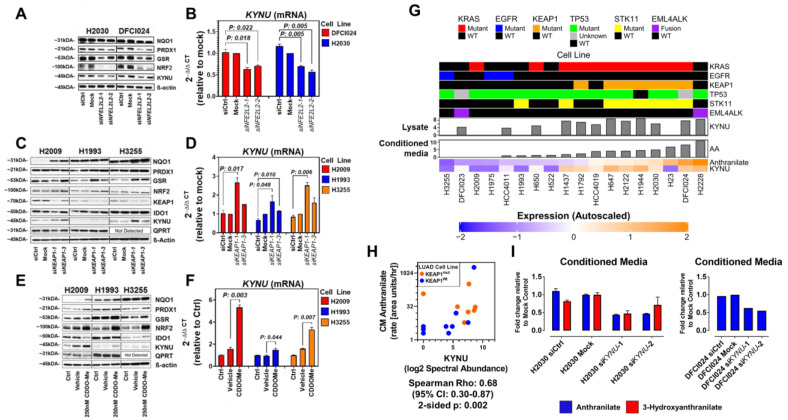
KYNU is regulated by NRF2 activation in lung adenocarcinomas and is functionally active. (**A**) Immunoblots for KYNU, NRF2 and NRF2-regulated enzymes peroxiredoxin 1 (PRDX1), glutathione reductase (GSR), and NAD(P)H quinone dehydrogenase 1 (NQO1) in *KEAP1/KRAS* mutant adenocarcinoma cell lines DFCI024 and H2030 following siRNA-mediated knockdown of the NRF2 transcribing gene *NFE2L2*. (**B**) mRNA expression (2^−ΔΔCt^) of *KYNU* following siRNA-mediated knockdown of *NFE2L2* in *KEAP1/KRAS* mutant adenocarcinoma cell lines DFCI024 and H2030. (**C**) Immunoblots for IDO1, KYNU, QPRT, KEAP1, NRF2, and NRF2-regulated enzymes PRDX1, GSR, and NQO1 following siRNA-mediated knockdown of *KEAP1* in H2009 (m*KRAS*), H1993 (*KRAS/EGFR* wild-type), and H3255 (m*EGFR*). (**D**) mRNA expression (2^−ΔΔCt^) of *KYNU* following siRNA-mediated knockdown of *KEAP1* in H2009, H1993, and H3255. (**E**) Immunoblots for IDO1, KYNU, QPRT, NRF2, and NRF2-regulated enzymes PRDX1, GSR, and NQO1 following 48-h treatment of H2009, H1993, and H3255 with NRF2-activator CDDOMe, vehicle (DMSO) or control media. (**F**) mRNA expression (2^−Ct^) of *KYNU* following 48-h treatment of H2009 (m*KRAS*), H1993 (*KRAS/EGFR* wild-type), and H3255 (m*EGFR*) with CDDOMe, vehicle or control media. CDDOMe: 2-Cyano-3,12-dioxo-oleana-1,9(11)-dien-28-oic acid methyl ester. (**G**) Heatmap illustrating the overall representation of whole lysate protein levels of KYNU and rates of anthranilate accumulation in conditioned media (CM) of 18 LUAD cell lines. Bar plots represent spectral abundance (log2) of KYNU and rate (log2 area units per hour per 500 μg of protein) of anthranilate accumulation in conditioned media. Columns were ranked from by rate of anthranilate accumulation in CM from lowest to highest. (**H**) Scatter plot illustrating the association between whole lysate protein levels of KYNU and rates of anthranilate accumulation in conditioned media (CM) of 18 LUAD cell lines. Nodes represent whether the respective cell line was wild-type (wt; blue) or mutant (mut; orange) for *KEAP1*. (**I**) Conditioned media abundance of anthranilate and 3-hydroxyanthranilate following siRNA-mediated knockdown of *KYNU* in *KEAP1/KRAS* mutant LUAD cell lines H2030 and DFCI024.

**Figure 3 cancers-14-02543-f003:**
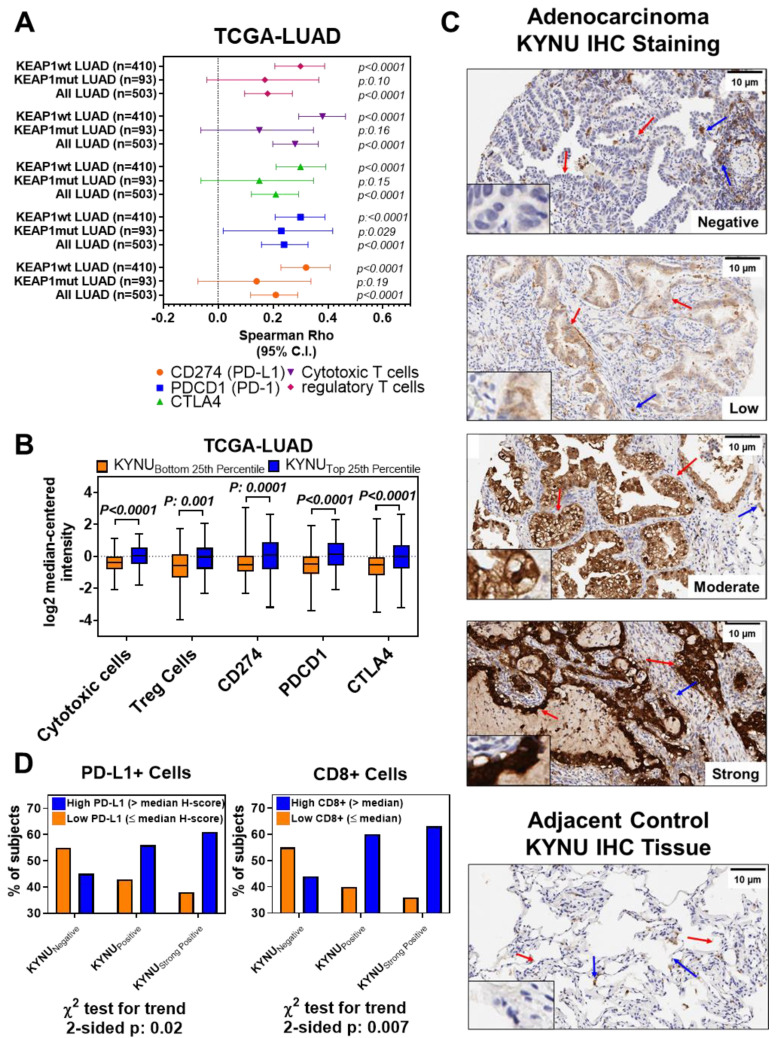
Association between KYNU and tumor immunophenotype. (**A**) Scatter plot illustrating Spearman correlations (95% CI) between continuous values of KYNU mRNA expression with gene-based signatures of cytotoxic T-cells, regulatory T cells, and immune checkpoint blockaded related genes CD274 (PD-L1), PDCD1 (PD1), and CTLA4 in TCGA-LUAD tumors stratified by presence or absence of KEAP1 mutations. (**B**) Distribution of gene-based signatures reflective of cytotoxic T-cells, regulatory T cells, and immune checkpoint blockade-related genes CD274 (PD-L1), PDCD1 (PD1), and CTLA4 in TCGA-LUAD tumors stratified by KYNU mRNA expression into the bottom and top 25th percentiles. Statistical significance was determined by a two-sided Wilcoxon rank sum test. (**C**) Representative immunohistochemistry (IHC) staining of KYNU protein expression in adenocarcinoma TMA stratified by KYNU protein expression (negative, positive (50–150) and strong positivity (>150)), as well as adjacent control tissue. Red arrows indicate tumor tissue while blue arrows indicate inflammatory cell staining. Insets show a 5× enlargement. (**D**) Association between KYNU protein expression (negative, positive (50–150) and strong positivity (>150)) and PD-L1 staining positivity and CD8+ TILs (E) in LUAD TMAs. PD-L1 was stratified based on whether PD-L1 was >median of the H-Score (high) or ≤median of H-scores (low). CD8+ levels were stratified into either high (>median of positive cells per millimeter (mm)^2^ tissue) or low (≤median of positive cells per mm^2^ tissue). Statistical significance was determined by two-sided *χ*^2^ test for trend.

**Figure 4 cancers-14-02543-f004:**
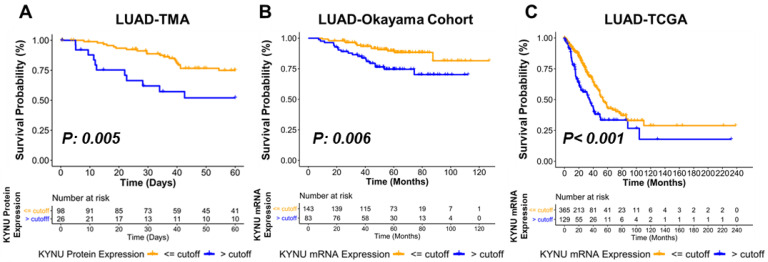
Association between KYNU and overall survival in LUAD. (**A**) Kaplan–Meier survival curves for five-year overall survival in adenocarcinoma TMAs based on a KYNU staining positivity cutoff of >120. Optimal cutoff value Figure 4. (**B**,**C**) Kaplan–Meier survival curves depicting overall survival in LUAD tumors stratified by an optimal cutoff value for *KYNU* mRNA Okayama and TCGA-LUAD gene expression datasets. Statistical significance was determined by two-sided log-rank Mantel–Cox test.

**Figure 5 cancers-14-02543-f005:**
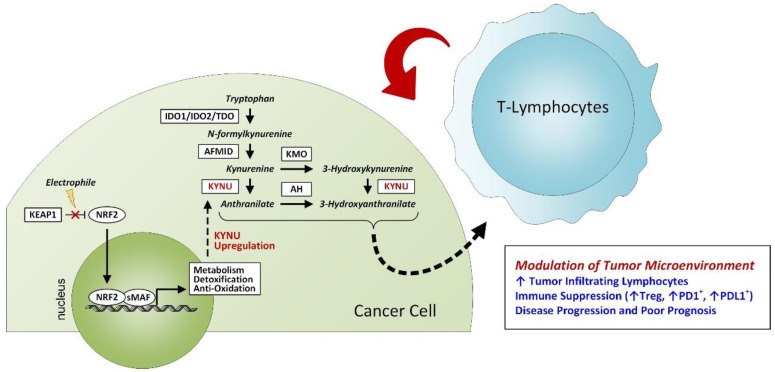
Proposed Schematic. NRF2 activation promotes KYNU upregulation in lung adenocarcinoma, resulting in an immunosuppressive tumor microenvironment characterized by elevated tumor T-cell infiltration, including T regulatory TILs, and concordant increases in protein expression of immune checkpoint blockade-related PD1 and PD-L1 and poor prognosis. Abbreviations: AFMID—arylformamidase; AH—anthranilate hydroxylase; IDO—indoleamine 2,3-dioxygenase; KEAP1—Kelch-like ECH-associated protein 1; KMO—kynurenine 3-monooxygenase; KYNU—kynureninase; NRF2—nuclear factor erythroid factor 2-related factor 2; PD1—programmed cell death protein-1; PDL1—programmed cell death ligand-1; sMAF—small musculoaponeurotic fibrosarcoma proteins; TDO—tryptophan 2,3-dioxygenase; Treg—regulatory T-cell.

**Table 1 cancers-14-02543-t001:** Cox proportional hazard models for KYNU protein expression and overall survival in lung adenocarcinoma TMA.

Variable	LUAD TMA					
	Univariable			Multivariable ^‡^		
	HR	95% C.I.	*p*	HR	95% C.I.	*p*
Sex						
Female	Reference			Reference		
Male	1.42	0.70–2.89	0.330	-		
Age ^¥^						
<65	Reference			Reference		
≥65	0.92	0.45–1.86	0.810	-		
Stage						
I	Reference			Reference		
II	3.69	1.43–9.53	0.007	4.12	1.59–10.69	0.004
III	7.74	2.98–20.10	<0.001	8.73	3.31–22.99	<0.001
IV	12.28	2.54–59.38	0.002	6.46	1.27–32.78	0.024
Smoking						
Never	Reference			Reference		
Former	1.11	0.36–3.40	0.860	-		
Current	1.23	0.40–3.73	0.720	-		
KYNU Staining ^†^						
≤Cutoff	Reference			Reference		
>Cutoff	2.77	1.33–5.80	0.007	3.11	1.41–6.84	0.005

^‡^ Variables included into the equation after selection using a backward stepwise method (likelihood ratio). ^†^ cutoff was defined as KYNU positivity > or ≤120. ^¥^ Stratified by median.

## Data Availability

Relevant data supporting the findings of this study are available within the Article and Appendices, or are available from the authors upon reasonable request.

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
