# Peer review of "Mutational Activation of the NRF2 Pathway Upregulates Kynureninase Resulting in Tumor Immunosuppression and Poor Outcome in Lung Adenocarcinoma"

_cancers, 2022, doi:10.3390/cancers14102543_

Round 1

Reviewer 1 Report

The article by Dr. Ostrin and the group indicated a novel mechanism of NRF2 tumoral immunosuppression through upregulation of KYNU. Though their findings are interesting, several things are required at this moment to be added to this manuscript before it is ready for acceptance. They are as follows:

  1. It has been known that arginine, cystine, glutamine, methionine, serine, and TRP were highly consumed AAs, regardless of the cell line or culture medium (Tohyama et al., 2016). Also, it's well known that oncogenic KRAS which is co-mutated with KEAP1 in lung cancer, upregulates several metabolic pathways including glutamine metabolism (PMID: 33870211). Now it will be worthwhile to discuss how and whether oncogenic KRAS also might play a role in tryptophan metabolism. The authors should add a few lines discussing this aspect in the discussion part. Since KRAS mutation occupies a decent number with KEAP1 mutation, it will be worthwhile to discuss as one of the future aspects of this manuscript.
  2. Authors should also mention whether there is any clinical trial ongoing based on biomarkers such as KRAS, and NFE2L2 mutations in lung cancers.
  3. The authors should add a model depicting the global message of this story.
  4. Since we know that NRF2 plays a significant role in glutamine metabolism (PMID: 31911550 and PMID: 28967864 and PMID: 28967920) and there are multiple links between glutamine and tryptophan metabolism (PMID: 27288072), any discussion on the theme of interlink between glutamine and tryptophan metabolism with the aspect of immune metabolism will be a worthwhile addition in the discussion part.

Author Response

Reviewer #1

The article by Dr. Ostrin and the group indicated a novel mechanism of NRF2 tumoral immunosuppression through upregulation of KYNU. Though their findings are interesting, several things are required at this moment to be added to this manuscript before it is ready for acceptance. They are as follows:

  1. It has been known that arginine, cystine, glutamine, methionine, serine, and TRP were highly consumed AAs, regardless of the cell line or culture medium (Tohyama et al., 2016). Also, it's well known that oncogenic KRAS which is co-mutated with KEAP1 in lung cancer, upregulates several metabolic pathways including glutamine metabolism (PMID: 33870211). Now it will be worthwhile to discuss how and whether oncogenic KRAS also might play a role in tryptophan metabolism. The authors should add a few lines discussing this aspect in the discussion part. Since KRAS mutation occupies a decent number with KEAP1 mutation, it will be worthwhile to discuss as one of the future aspects of this manuscript.

Response: We have now expanded the discussion section of the revise manuscript per the Reviewer’s recommendation.

  1. Authors should also mention whether there is any clinical trial ongoing based on biomarkers such as KRAS, and NFE2L2 mutations in lung cancers.

Response: We have elaborated on ongoing clinical trials in the context of KRAS/NRF2 lung cancers in the discussion section of the revised manuscript.

  1. The authors should add a model depicting the global message of this story.

Response: We have now added a schematic describing the seminal findings of our study.

  1. Since we know that NRF2 plays a significant role in glutamine metabolism (PMID: 31911550 and PMID: 28967864 and PMID: 28967920) and there are multiple links between glutamine and tryptophan metabolism (PMID: 27288072), any discussion on the theme of interlink between glutamine and tryptophan metabolism with the aspect of immune metabolism will be a worthwhile addition in the discussion part.

Response: We have now included content regarding glutamine and tryptophan metabolism in the context of immune cells and immune cell effector functions in the discussion section of the revised manuscript.

Reviewer 2 Report

In this manuscript authors showed that NRF2 can alter tryptophan metabolism through the kynurenine pathway increasing KYNU expression proving that NRF2 is a regulator of KYNU expression in LUAD. Moreover, increased KYNU was associated with immunosuppression, including potent induction of T-regulatory cells, increased levels of PD1 and PD-L1, and poorer survival. 

The manuscript is clear and generally well written. However, some points must be improved. In particular:

  • The introduction section provides a limited background of the topic. Authors should expand this section underlying that Nrf2 is considered a master regulator of oxidative stress response (PMID: 33123312) and  that oxidative stress has a key role in the progression of several solid tumors (PMID: 35453297; PMID: 34638427). A short introduction on KYNU function is also necessary. Importantly, authors need to specify that NRF2 pathway plays a key role in other cancers such as pancreatic (PMID: 35052602) and ovarian cancer (PMID: 35453348). This is an important point since it can explain the multifaceted action of this signaling suggesting that results found by the authors could also be valid in other type of cancer. 

  • "2.3. Western Blot Analysis ": antibodies dilutions must be reported
  • Figure 3C. Scale bar and negative control must be shown. Higher magnification images for each KYNU staining (as inserts or separate images) would be appreciated.  
  • an accurate revision of punctuation is recommended

Author Response

Reviewer #2

In this manuscript authors showed that NRF2 can alter tryptophan metabolism through the kynurenine pathway increasing KYNU expression proving that NRF2 is a regulator of KYNU expression in LUAD. Moreover, increased KYNU was associated with immunosuppression, including potent induction of T-regulatory cells, increased levels of PD1 and PD-L1, and poorer survival. 

The manuscript is clear and generally well written. However, some points must be improved. In particular:

  • The introduction section provides a limited background of the topic. Authors should expand this section underlying that Nrf2 is considered a master regulator of oxidative stress response (PMID: 33123312) and that oxidative stress has a key role in the progression of several solid tumors (PMID: 35453297; PMID: 34638427). A short introduction on KYNU function is also necessary. Importantly, authors need to specify that NRF2 pathway plays a key role in other cancers such as pancreatic (PMID: 35052602) and ovarian cancer (PMID: 35453348). This is an important point since it can explain the multifaceted action of this signaling suggesting that results found by the authors could also be valid in other type of cancer. 

Response: We have now provided additional content on NRF2 as a master anti-oxidant regulator in the broader context of cancer as well as introducing the function of KYNU in the introduction section of the revised manuscript regarding. 

  • "2.3. Western Blot Analysis ": antibodies dilutions must be reported

Response: Antibodies dilution have now been added.

  • Figure 3C. Scale bar and negative control must be shown. Higher magnification images for each KYNU staining (as inserts or separate images) would be appreciated.  

Response: We have now added scale bars and negative controls for the respective IHC sections. High magnification images have also been included. As an external positive control for KYNU staining, we used non-neoplastic liver tissue stained with KYNU antibody where as we substituted the KYNU antibody with diluent for negative control. This information is now provided in Figure A5 in the Appendix.

  • an accurate revision of punctuation is recommended

Response: We have gone through the revised manuscript thoroughly to ensure correct use of punctuation.

Round 2

Reviewer 1 Report

All concerns have been addressed, ready for acceptance.